# Distinct Classes of Flavonoids and Epigallocatechin Gallate, Polyphenol Affects an Oncogenic Mutant p53 Protein, Cell Growth and Invasion in a TNBC Breast Cancer Cell Line

**DOI:** 10.3390/cells10040797

**Published:** 2021-04-02

**Authors:** Madhu Kollareddy, Luis A. Martinez

**Affiliations:** 1Cancer Epigenetics Laboratory, School of Cellular and Molecular Medicine, University of Bristol, Bristol BS8 1TD, UK; 2Stony Brook Cancer Center, Department of Pathology, Renaissance School of Medicine, Stony Brook University, Stony Brook, NY 11790, USA

**Keywords:** mutant p53, ETS, gain-of-function, flavonoids, polyphenol, TNBC, breast cancer, nucleotide metabolism genes, DNA damage, replication stress, metastasis

## Abstract

Mutant p53(s) are widely considered as oncogenes and promote several gain-of-function oncogenic activities. p53 mutations correlate with higher rates of metastasis and poor survival; therefore, it is paramount to inhibit mutant p53 protein either directly or indirectly. Although some compounds have been developed, none of them have achieved a desirable level of specificity. Some of these compounds only targeted specific mutations. In search of less-toxic compounds, we tested plant-derived compounds on mutant p53 triple-negative breast cancer cell lines. Here, we show that the compounds tested reduced the protein levels of one of the more frequent oncogenic p53 mutants (R249S; hot spot mutation), and its important targets that promote invasion and metastasis, including GMPS and IMPDH1. All compounds tested perturbed the invasion potential of the breast cancer cell line. These compounds downregulated several nucleotide metabolism genes (NMGs) which are essential for cell cycle progression. We observed S-phase arrest correlating to reduced cell proliferation and increased replication stress. Moreover, we also show a reduction of key ETS transcription family members including ETS2, ETS1, ETV1, and ETV4, which are involved in invasion and metastasis. We propose that these compounds may inhibit invasion by interfering with multiple pathways. Our findings exemplify that these tested compounds could inhibit invasion and cell growth in TNBC in a nucleotide-dependent manner.

## 1. Introduction

Roughly 50% of all human cancers harbor several different gain-of-function mutations [1]. A single point mutation in wild-type p53 converts a tumor-suppressive protein into an oncogenic gain-of-function mutant, with wild-type and mutant p53 (mtp53) having opposing functions; wild-type p53 suppresses cancer development, whereas mtp53 promotes cancer by acting with other oncogenes. Mtp53 promotes aggressive metastasis, which correlates with reduced overall survival [2]. mtp53 promotes several oncogenic activities, including proliferation, drug resistance, metabolic alterations, invasion, and metastasis by regulating the expression of several genes [3]. Since it is a transcription factor, mtp53 is difficult to target with small-molecule inhibitors due to its high disorderliness and dynamic nature. Few drugs have shown to be effective in restoring wild-type p53 functions of mp53 in cell line and xenograft models [4,5]. Unfortunately, none of these compounds progressed into clinical trials due to specificity and toxicity issues. Hence, there is an urgent need to identify and characterise less-toxic compounds that are specific to mtp53 cancers.

In an unbiased approach, we tested specific plant-derived natural compounds to assess their potential to inhibit cell growth arrest and invasion in mtp53 cancer cell lines. The effect of these compounds on mtp53 protein levels is not known. We screened distinct classes of flavonoids (apigenin, genistein, quercetin), polyphenol (epigallocatechin gallate, EGCG), phenol (silymarin), and indole (diindolylmethane). Among the non-toxic flavonoids found ubiquitously in plants, apigenin was shown to have promising anticancer properties. Wei et al. showed that apigenin inhibited skin tumorigenesis induced by DMBA and TPA in SENCAR mice [6]. Recently it was shown that apigenin can restore wild-type activity of mtp53 in pancreatic cancer cells [7]. EGCG is one of the important and potent antioxidants found in green tea [8]. It was shown to be effective in a mtp53 TNBC (triple-negative breast cancer) cell line model [9]. It specifically upregulated pro-apoptotic genes and inhibited pro-survival genes. Moreover, EGCG was shown to be potent in cancer cell lines, but not in normal fibroblast WI-38 [10]. EGCG is currently under evaluation in several clinical trials for the treatment of several cancers [11,12]. Genistein, an isoflavone found in soy products, was shown to target multiple oncogenic signaling pathways in cancers [13]. Matsukawa et al. showed that genistein inhibited gastric cancer cell growth by arresting cells at G_2_/M [14]. *BRCA1* mutant breast cells were highly sensitive to genistein; it induced a DNA damage response, mitotic catastrophe, and polyploidy [15]. The importance of quercetin, a flavonol found in a variety of vegetables and fruits, was realized in 1950, with several anticancer and anti-inflammatory properties being attributed to quercetin [16]. In line with our study, it was shown to reduce mtp53 protein levels in the MDA-MB-468 cell line in a dose- and time-dependent manner [17]. Silymarin is a flavonolignan found in milk thistle (*Silybum marianum*). Data from several reports suggest that it has an important role in cancer chemoprevention [18]. 3,3′-Diindolylmethane (DIM) is a glucosinolate widely found in cruciferous vegetables [19]. It was proven to have pleiotropic effects on multiple signaling pathways. DIM induced p21 levels and associated G_1_ arrest both in p53 wild-type and p53 mutated breast cancer cell lines [20]. DIM inhibited invasion and tube formation in HUVEC cells. Under in vivo settings, it reduced neovascularization by 72% [21]. Several clinical trial studies were completed, and some are underway for DIM, particularly aimed at prostate cancer.

In our study, mtp53 breast cancer cell lines displayed differential sensitivities to the above-described compounds. This was particularly true for the hotspot p53 mutation R249S containing the cell line BT-549, which was relatively more sensitive to all compounds. Intriguingly, these compounds induced the downregulation of mtp53 and its binding partner ETS2 [22]. The downregulation of mtp53 affected cell cycle distribution and severely perturbed invasion. Moreover, a reduction in mtp53 levels corresponded to the downregulation of its direct transcriptional targets—namely, the nucleotide metabolism genes [23]. In our in vitro study, these compounds reduced the invasive potential of BT-549, which was derived from a TNBC patient. This is the first study of its kind showing the potential of plant-derived compounds to downregulate mtp53, its binding partner ETS2, and their direct transcriptional targets in a TNBC cell line.

## 2. Materials and Methods

### 2.1. Cell Lines and Plant-Derived Compounds

BT5-49, HCC38, and MDA-MB-231 were purchased from ATCC and cultured according to the vendor’s instructions. All cell lines were free of mycoplasma. Apigenin, EGCG, genistein, quercetin, silymarin, and 3,3′-Diindolylmethane were purchased from Sigma (St. Louis, MO, USA). Stocks of 100 mM apigenin, EGCG, genistein, quercetin, silymarin, and 3,3′-Diindolylmethane were prepared using DMSO.

### 2.2. Cell Proliferation Assay

Ten thousand were seeded in 96-well plates in 80 μL of respective media and incubated overnight. Several drug concentrations were prepared by serial dilutions and 20 μL (5×) of each drug concentration was added in triplicates. After 72 h of incubation, 10 μL of MTT (5 mg/mL) (Boston bioproducts, Ashland, MA, USA) was added. Cells were incubated in a tissue culture incubator for 3 h, and finally, 50 μL of SDS lysis buffer (10% SDS pH 5.5, 0.01 M HCl) was added and incubated overnight. Plates were read at 562 nm using a Synergy plate reader. Percentage survival was calculated relative to control (100% survival). All MTT assays were performed in three biological replicates, with each containing three technical replicates.

### 2.3. Lentiviral-Mediated Expression of R249S Mutant

Lentivirus-mediated expression of R249S mutant was carried out as previously described [23]. Briefly, the p53 mutation was introduced into the p53 coding sequence for the R249S (Q5 Site-Directed Mutagenesis kit, New England Biolabs, Ipswich, MA, USA), then the cDNAs were cloned into the lentiviral vector pLVX-Hygromycin (Clontech, Mountain View, CA, USA), and lentiviruses were used to infect WI-38. The cells were selected with hygromycin for one week.

### 2.4. Western Blotting

An amount of 0.3 × 10^6^ cells were seeded in 6-well plates and incubated overnight. Cells were treated for 24 h with 1/2× and 1× IC_50_ concentrations. Whole-cell lysates were prepared using RIPA buffer supplemented with protease and phosphatase inhibitors. Lysates were sonicated and centrifuged at 20,000× *g* at 4 °C to remove insoluble material. Protein concentration was determined using a Micro BCA Protein Assay kit (Thermo Fisher Scientific, Waltham, MA, USA) and equal amounts of protein were resolved using NuPAGE Bis-Tris gels (Thermo Fisher Scientific, Waltham, MA, USA). Western blot analysis was performed using DCK (GeneTex, Irvine, CA, USA: GTX107636; 1:2000), RRM2 (GeneTex: GTX103193; 1:2000), TK1 (GeneTex: GTX62133; 1:2000), RRM1 (Cell signaling, Beverly, MA, USA: 8637S; 1:3000), TYMS (Cell signaling: 9045P; 1:3000), DTYMK (Protein Tech, Rosemont, IL, USA: 15360-1-AP; 1:3000), DHFR (Protein Tech:15194-1-AP; 1:3000), IMPDH1 (Protein Tech: 22092-1-AP; 1:3000), GMPS (Santa Cruz, Dallas, TX, USA: sc-376163; 1:3000), p53 (Santa Cruz: sc-126; 1:3000), ETS2 (Santa Cruz: sc-365666; 1:200), γH2AX (Bethyl: A300-081A; 1:2000), and actin HRP (Sigma, St. Louis, MO, USA: A3854; 1:40,000) antibodies.

### 2.5. qRT-PCR

RNA was isolated using TRIzol reagent (Molecular Research Center, Inc., Cincinatti, OH, USA) according to the manufacturer’s instructions. RNA integrity was verified using an RNA chip (Agilent bioanalyzer, Santa Clara, CA, USA). One microgram of RNA was used to synthesize cDNA (20 µL reaction) (Bio-Rad’s Iscript select cDNA synthesis kit) according to the manufacturer’s protocol. The cDNA was diluted 1:10 and used for qRT-PCR with SYBR green detection (Invitrogen, Waltham, MA, USA). Phusion HF reaction buffer, Phusion HF DNA polymerase, and dNTPs used for qRT-PCR were purchased from New England Biolabs. Bio-Rad CFX96 Touch real-time PCR detection system was also used. Biorad CFX manager software was used to calculate fold changes in the gene expression. *HPRT* housekeeping gene was used as an internal reference for normalisation. All primer sequences are listed in Appendix A. All nucleotide metabolism genes (NMGs) are defined in Appendix A.

### 2.6. Cell Count and Doubling Time Determination

An amount of 100,000 cells were seeded in 6-well plates, and after 6 h, cells were treated with 1/2× and 1× IC_50_ concentrations. After 72 h, cells were trypsinised and harvested for cell counting. A Beckman cell counter was used to determine cell density. The equation X = (logb − loga)/0.301 was used to calculate population doubling times.

### 2.7. Cell Cycle Analysis

An amount of 0.5 × 10^6^ cells were seeded in 60 mm dishes and incubated overnight. Cells were treated with 1× and 2× IC_50_ concentrations for 24 h. The media was aspirated and washed once with PBS and trypsinised. Cells were collected, centrifuged at 500× *g* for 5 min at 4 °C, and washed with PBS. Finally, cells were fixed in 1 mL of ice-cold 70% ethanol and stored at −20 °C until analysis. After overnight fixation, cells were washed with PBS and treated with RNase (10 mg/mL) for 15 min followed by staining with propidium iodide (Sigma) (50 μg/mL). Stained cells were analyzed using the Beckman coulter flow cytometer (488 Argon laser). Event counts were 10,000 cells per sample.

### 2.8. Invasion Assay

Invasion assay was carried out according to the manufacturer’s instructions (BD BioCoat 24-well Matrigel invasion chambers). Invasion of treated cells was expressed as a percentage relative to the control cells.

### 2.9. Statistical Analysis

All statistical analyses were performed using Microsoft Excel software. Statistical analysis was performed using a two-tailed paired Students’s *t*-test to evaluate the significance of the differences between the groups. *p*-values < 0.05 (two-tailed) were considered statistically significant. *, **, and *** denote *p* < 0.05, *p* < 0.01, and *p* < 0.001, respectively. “Ns” means not significant.

## 3. Results

### 3.1. mtp53 Cell Lines Displayed Differential Sensitivities, Compounds Reduced Cell Growth and Affected Cell Cycle Distribution

Initially, we screened all compounds on three different mtp53 human breast cancer cell lines, BT-549 (R249S), HCC38 (R273L), and MDA-MB-231 (R280K). All three were triple-negative breast tumor cell lines, where few options are available for chemotherapeutic intervention. Among the cell lines, BT-549 showed high sensitivity to apigenin, EGCG, genistein, quercetin, silymarin, and 3,3’-Diindolylmethane (Figure 1A). HCC338 was the second most sensitive cell line to all the compounds except silymarin and 3,3’-Diindolylmethane. We anticipated resistance in the MDA-MB-231 cell line, as this cell line is inherently drug-resistant. IC_50_ values for all compounds are shown in Appendix A. We characterized the mechanism of action of these compounds on BT-549, which is the most sensitive cell line. Cell proliferation rate was decreased >2 fold for all compounds at 1× IC_50_ (Figure 1B). Particularly, EGCG severely perturbed cell growth. Consequently, doubling times also increased, which are noted above each bar (Figure 1B). Parallel to this, we performed flow-cytometry-based cell cycle analysis. At 1× IC_50_ concentration, apigenin, EGCG, genistein, quercetin, and silymarin induced S-phase arrest (Figure 1C,D). 3,3′-Diindolylmethane induced S-phase arrest at 2× IC_50_. An accumulation of G_2_/M at 1× IC_50_ was noticed for apigenin, genistein, and silymarin. At 2× IC_50_, a small increase in sub G_1_ (apoptotic) was evident for all compounds (Figure 1C).

### 3.2. Plant-Derived Compounds Reduced mtp53 Protein Levels

Next, we assessed the effect of these compounds on mtp53 protein levels. For this study, we selected BT-549 as this represents the most sensitive cell line. We treated cells with 1/2× and 1× IC_50_ concentrations for 24 h. Interestingly, mtp53 levels were reduced at 1/2× IC_50_ concentrations of apigenin, genistein, and quercetin (Figure 2A,B). However, the downregulation of mtp53 by EGCG and silymarin was apparent at 1× IC_50_ concentration. 3,3’-Diindolylmethane showed only a modest effect on mtp53 protein levels. We and others have shown mtp53 associates with other transcription factors to mediate its gain-of-function oncogenic activities. Particularly, mtp53 associates robustly with ETS2 to regulate transcription [5]. We questioned if these compounds have any effect on ETS2. All compounds except quercetin reduced the protein levels of ETS2 (Figure 2A). Since mtp53 protein levels were reduced upon treatment, we checked the protein levels of its key direct transcriptional targets. Almost all proteins tested were reduced at 1× IC_50_. For some compounds, 1/2× IC_50_ was sufficient to show this effect. Particularly, invasion- and metastasis-promoting enzymes GMPS (part of MammaPrint 70-gene signature) and IMPDH1 were reduced. Likewise, rate-limiting enzymes RRM1 and RRM2 (both catalyzing the reduction of ribonucleotides to deoxyribonucleotides) were reduced. Apigenin did not reduce TK1 protein levels, but rather we saw upregulation at 1× IC_50_. In summary, most of the compounds reduced mtp53 protein levels and its direct transcriptional targets. We also probed for cleaved PARP, which is an apoptotic marker. There was only a modest increase of cleaved PARP with apigenin, EGCG, and quercetin (Figure 1A). Mechanistically, S-phase arrest (Figure 1C,D) can be explained by the substantial reduction of NMGs. We have previously shown that the reduction of NMGs correlates with a reduced nucleotide pool. Hence, S-phase arrest could be potentially due to the imbalance in intracellular nucleotide pools [23]. Since these compounds affected NMG expression, we believe that these compounds therefore have the potential to induce replication stress. Indeed, there was an induction of γH2AX, a putative marker of replication stress and DNA damage response (Figure 2A). Taken together, our data suggest that these compounds can induce replication stress and DNA damage response by reducing protein levels for the key proteins involved in cell cycle progression.

We also examined the influence of these compounds on a non-cancerous cell line, WI-38, which contains wild-type p53. Apigenin and quercetin induced protein levels of wild-type p53, contrary to the mutant p53 cell line BT-549 (Appendix A). In the wild-type cell line, NMG protein levels did not change substantially, except for those of RRM2 and TK1. Interestingly, the introduction of mtp53 (R249S) into the WI-38 cell line upregulated all NMG protein levels compared to control cells. When R249S was stably expressed, the effect of these compounds on NMGs was much more profound when compared to the control vector cells. Concurrent expression of R249S mutant and treatments induced enhanced apoptotic cell death (cleaved PARP) compared to the control (Appendix A).

Next, we asked whether these compounds induce the degradation of mtp53 through the ubiquitin-proteasome-dependent pathway (UPP). For this, we used MG132, which inhibits UPP. Only genistein induced a reduction of mtp53, which was moderately rescued when co-treated with MG132, suggesting that genistein induces the degradation of mtp53 via UPP (Figure 2C,D).

### 3.3. Direct Transcriptional Targets of Mutant p53 Were Affected

Since all the compounds modulated the protein levels of mtp53 and its targets, we did qRT-PCR to determine the effect on NMG transcripts. EGCG and genistein reduced mtp53 RNA levels in a dose-dependent manner (Figure 3). A similar pattern was noticed for its partner ETS2 and most of its targets. Silymarin downregulated the expression of mtp53 and ETS2 at the protein level, however, it did not affect *mtp53* and *ETS2* transcripts. This indicates that silymarin could modulate mtp53 and ETS2 stability at the protein level. Nonetheless, we saw a reduction of NMG transcripts in a dose-dependent manner. 3,3′-Diindolylmethane modestly affected *mtp53* transcript and this was also true at the protein level. NMG transcripts were affected at the highest concentration (1× IC_50_). In summary, EGCG, genistein, and to some extent 3,3′-Diindolylmethane, also downregulated *mtp53* and its targets transcriptionally, which correlates with protein levels. Only silymarin seemed to affect mtp53 stability at the protein level.

Next, we asked if these compounds have any effect on other key ETS transcription factor family members. EGCG and silymarin induced the downregulation of *ETS1*, *ETV1*, and *ETV4* genes. EGCG specifically affected *ETV5*, whereas Silymarin affected *ERG* at ½× IC_50_ (Figure 4, Appendix A). Genistein was able to modestly downregulate *ETS1*, *ETV1*, and *ETV5* at 1× IC_50_ (Figure 4). 3,3′-Diindolylmethane was effective in reducing *ETV1* and *ETV4* transcripts (Figure 4), with both shown to promote Ewing’s sarcoma. We did not see a substantial downregulation of several other ETS family members, including *ELF1*, *ESE1*, *ESE2*, *Fli1*, *PDEF*, and *ERG* (Appendix A).

### 3.4. Treatment with Natural Compounds Perturbed Invasion

Several signal transduction pathways, including Ras/MAPK and Rho GTPases mediated migration and invasion, are regulated by GTP [23]. All compounds significantly downregulated IMPDH1, a rate-limiting enzyme in GTP synthesis, as well as GMPS, involved in the final step of synthesis of guanosine monophosphate (Figure 2A). We determined the invasive potential of BT-549 in the presence of these compounds. Indeed, a dramatic inhibition of the invasion of BT-549 at 1× IC_50_ was clearly evident with apigenin, EGCG, quercetin, silymarin, and 3,3′-Diindolylmethane (Figure 5).

## 4. Discussion

Inhibiting mutant p53 functions, whilst at the same time restoring its wild-type functions, is an active area of research. Towards this aim, several compounds have been designed and tested. However, these compounds are not specific to mutant p53(s), and our unpublished data show that they are also active on wild-type p53 cell lines. Hence, the main problem is the lack of specificity towards mutant p53. Moreover, these compounds are designed only to target specific p53 mutations. An ideal approach would be to develop compounds that can downregulate mutant p53 levels. We thought the best approach was to screen plant-derived compounds present in many vegetables and fruits that we find regularly in our diet. Our approach is unbiased, as there is no prior knowledge on the effectiveness of these compounds on mutant p53. Strikingly, our report demonstrates that the tested compounds were able to reduce mutant p53 levels and its direct transcriptional targets. One could speculate that reduced mutant p53 levels could be due to cell cycle effects, but we ruled out this possibility as we showed in our previous study that mutant p53 levels are not cell-cycle regulated. We specifically screened on TNBC mtp53 cells, as there are only a few therapeutic options available. Moreover, this sub-type of breast cancer is very resistant to anticancer drugs [24]. TNBC cell lines displayed differential sensitivities to the compounds. This is expected because these cell lines have different p53 mutations, differential genotypes, and may express different levels of drug transporters. For example, MDA-MB-231 harbors oncogenic mutations in *K-Ras* and *B-Raf*, whereas BT-549 has a *PTEN* mutation [25]. These factors, and other unknown considerations, may contribute to differential sensitivities. Nevertheless, we noticed that these compounds induced a typical phenotype similar to mtp53 knockdown (reduced proliferation and invasion). Apparently, most of the compounds reduced the mRNA expression of *mtp53*, and this matches protein levels. However, genistein seems to regulate mtp53, both RNA and protein levels, as treatment of cells with the proteasome inhibitor MG132 in the presence of genistein rescued mtp53 protein levels. This observation agrees with a study reported by Li et al. where genistein downregulated MDM2 at both RNA and protein levels [26]. Additional studies to identify ubiquitin ligases involved in genistein-mediated degradation of mtp53 would be useful to gain new insights into mtp53 protein stability. On the other hand, whilst silymarin minimally affected RNA levels of mtp53, it downregulated mtp53 protein level, suggesting that it potentially regulates mtp53 protein stability. Finally, DIM had an opposing effect, where it reduced mtp53 RNA levels, but protein levels were minimally affected. Even so, mtp53’s direct targets, NMGs which promote various oncogenic activities, were significantly downregulated in the mtp53 TNBC cell line.

The ETS transcription factor family regulates gene expression in cancer cell lines. In general, they are regarded as oncogenes, as they are involved in promoting general oncogenic phenotypes including proliferation, metastasis [27], and drug resistance [22,28,29]. Ewing’s sarcoma typically contains EWS-FLI1 fused oncogenic protein. Inhibition of this fused protein with a small-molecule inhibitor stalled cell growth [30]. Moreover, targeting ETV1 and ERG in prostate cancers significantly reduced invasion and metastasis both in vitro and in vivo [31,32]. Our tested plant-derived compounds downregulated key ETS transcription members including *ETS2*, *ETS1*, *ETV1*, *ETV4*, and *ETV5*. Testing of these compounds in Ewing’s sarcoma and prostate cancer is worthwhile as these cancers are characteristic of aberrant ETS genetic lesions.

Targeting cell cycle progression and activation of checkpoint pathways or DNA damage response has become an important strategy in recent years to induce cytotoxic and cytostatic effects. Many small-molecule inhibitors were developed to this end. A few examples include CDK [33], Aurora kinases [34], and polo-like kinase inhibitors [35]. Targeting CDK, Aurora kinases, and Polo-like inhibitors elicited a robust cell cycle checkpoint activation and DNA damage response. Most of the natural compounds we used induced a similar phenotype including either G_1_ or S or G_2_/M arrest, and we observed some apoptotic cell death even at low concentrations. One of the major problems reported with synthetic cell cycle inhibitors is inevitable cytotoxicity at various levels including neutropenia, anemia, mucositis, and pneumonia [36]. The compounds we used were proven to be less toxic and naturally available. It is apparent that these compounds evoked cell cycle checkpoint and DNA damage response through inhibiting NMGs. Upregulation of γH2AX also correlated to a reduction in DCK enzyme. We have previously shown that DCK, a salvage pathway enzyme, is involved in the recycling of degraded cytidine [23]. Cancer cells have high levels of dTTP relative to non-cancerous cells. High levels of dTTP inhibit ribonucleotide reductase enzymes, thereby reducing cytidine synthesis [23]. To maintain this, DCK compensates the homeostasis of cytidine, hence a further reduction of DCK proteins is potentially linked to replication stress. Usage of these compounds may potentially halt cancer cell cycle progression with manageable toxicities. This is supported by our data showing that these compounds displayed opposite effects in a non-cancerous cell line (i.e., upregulation of wild-type p53). These plant-derived compounds could have a greater therapeutic window. Our study potentially opens the door for in-depth characterisation of these compounds, including the mechanistic basis of mtp53 downregulation. Synthesis of derivatives (structure–activity relationship studies) from these lead compounds could further aid in increasing specificity and potency for mtp53 cancers.

## Figures and Tables

**Figure 1 cells-10-00797-f001:**
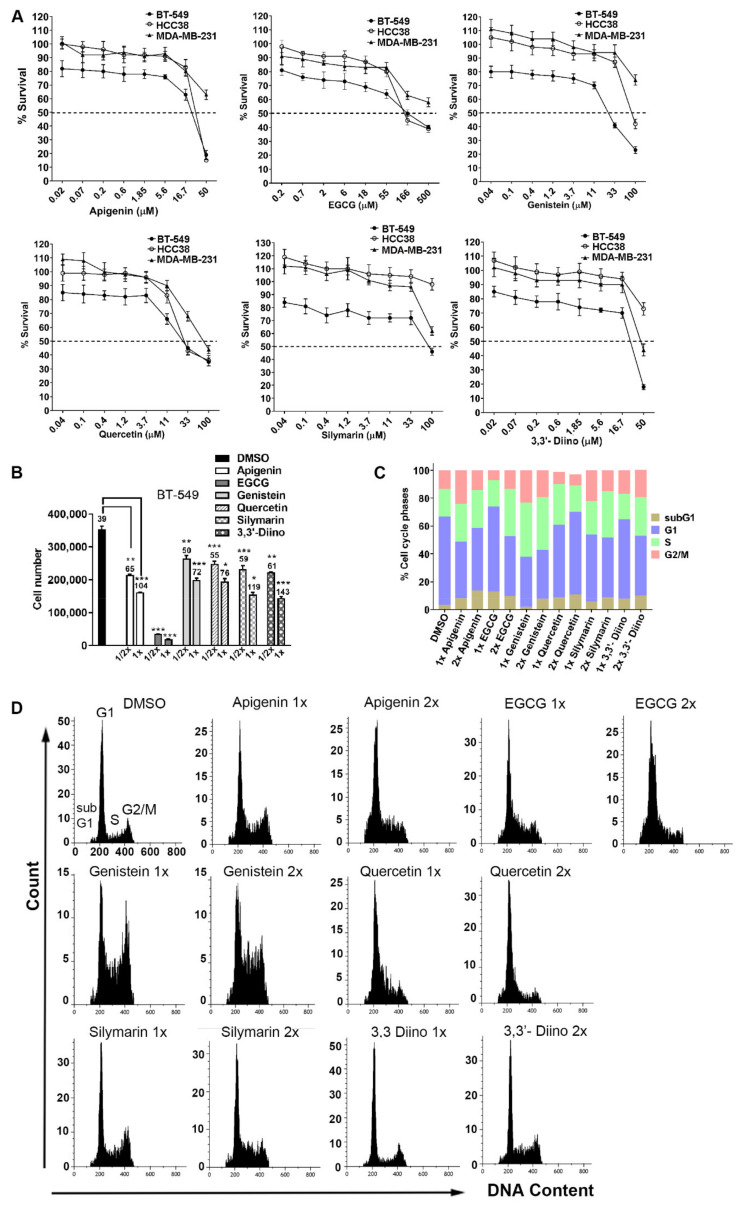
Screening of natural compounds on the mutant p53 cell lines. (**A**) MTT-based dose–reScheme 549. HCC38, and MDA-MB-231 cells were treated with plant-derived compounds in serial dilutions. MTT assays were performed in three biological replicates (*n* = 3). Error bars represent mean ± SD derived from three independent replicates. (**B**) BT-549 cells were treated with 1/2× and 1× IC_50_ concentrations to evaluate the effect on cell proliferation. After 48 h, total cell density was determined to calculate population doubling times. Numbers above the graph represent doubling times in hours. Error bars represent mean ± SD calculated from three independent experiments. *, **, and *** denote *p* < 0.05, *p* < 0.01, and *p* < 0.001, respectively. NS: not significant. (**C**) Bar graphs representing the quantification of various phases of the cell cycle in BT-549 after treatment with either DMSO or indicated compound for 24 h at 1× and 2× IC_50_ values. (**D**) Corresponding cell cycle histograms for BT-549 as treated above. Cells were fixed in 70% ethanol overnight before performing propidium iodide-based flow cytometry cycle analysis.

**Figure 2 cells-10-00797-f002:**
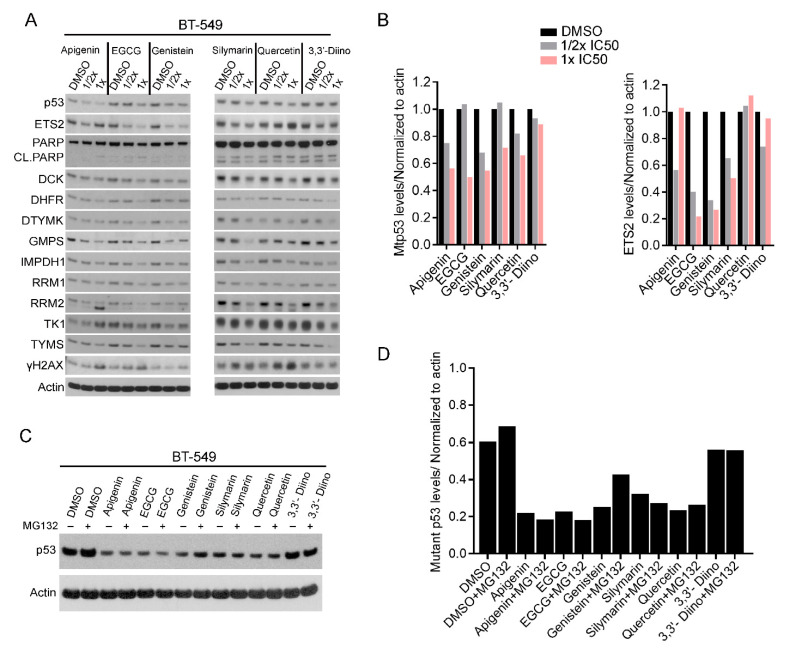
Plant-derived compounds reduced mutant p53, its bind partner, and its direct targets at the protein level and induced replication stress. (**A**) BT-549 cells were treated with either DMSO or plant-derived compounds at 1/2× and 1× IC_50_ concentrations for 24 h. Cells were lysed and processed for Western blot analysis. Actin was used as a loading control. (**B**) Bar graphs show densitometric analysis of p53 protein levels after normalization to actin. (**C**) BT-549 was treated with 1× IC_50_ for 24 h and 4 h prior to harvesting cells for protein lysates, MG132 (10 µM) was added. (**D**) The bar graphs represent densitometric analysis of p53 in the presence or absence of MG132 after normalization to actin.

**Figure 3 cells-10-00797-f003:**
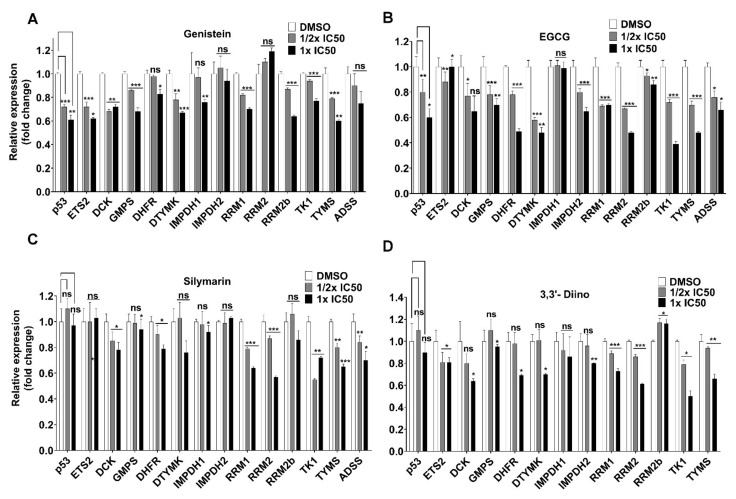
The effect of compounds on mtp53 transcript and its targets. BT-549 cells were treated with 1/2× and 1× IC_50_ concentrations for 24 h. RNA was extracted to prepare cDNA and qRT-PCR was performed for nucleotide metabolism genes. *HPRT* was used as a reference housekeeping gene for normalisation. Here, the dose-dependent reduction of *mtp53*, *ETS2*, and nucleotide metabolism gene transcripts can be seen after the treatment with EGCG (**A**), Genistein (**B**), Silymarin (**C**), and 3,3’-Diino (**D**). Errors bars represent mean ± SD derived from three independent replicates. *, **, and *** denote *p* < 0.05, *p* < 0.01, and *p* < 0.001, respectively. “ns” means not significant.

**Figure 4 cells-10-00797-f004:**
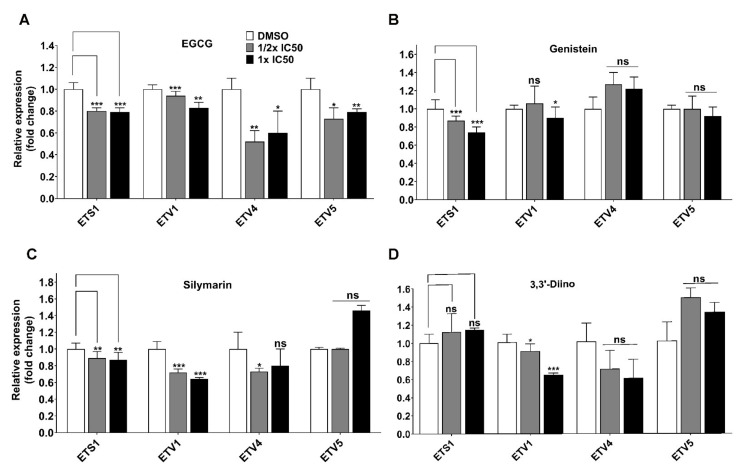
The modulation of key ETS members in response to natural compounds. BT-549 cells were treated with 1/2× and 1× IC_50_ concentrations for 24 h. RNA was extracted to prepare cDNA and qRT-PCR was performed for key ETS family members. After the treatment with EGCG (**A**), Genistein (**B**), Silymarin (**C**), and 3,3’-Diino (**D**) modulation of ETS members can be seen. *HPRT* was used as a reference housekeeping gene for normalization. Error bars represent mean ± SD derived from three independent replicates. *, **, and *** denote *p* < 0.05, *p* < 0.01, and *p* < 0.001, respectively. “ns” means not significant.

**Figure 5 cells-10-00797-f005:**
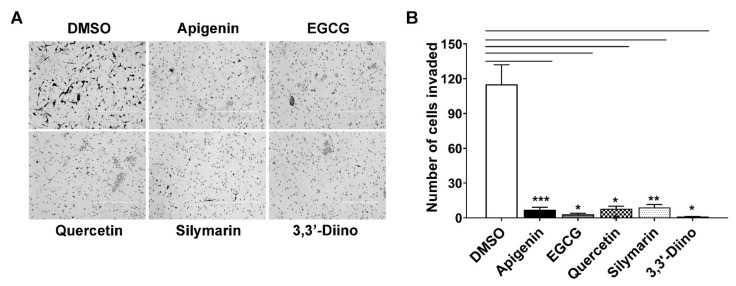
Plant-derived compounds significantly reduced the invasion of BT-549. (**A**) BT-549 cells were seeded in the upper side of the Matrigel invasion chamber in the media without fetal bovine serum. The lower chamber was filled with complete media. After 24 h, the number of cells that had invaded the lower chamber was counted at five different places. Scale bars, 400 µm. (**B**) Quantification of the number of invaded cells relative to the DMSO control. Error bars represent mean ± SD calculated from three independent experiments. *, **, and *** denote *p* < 0.05, *p* < 0.01, and *p* < 0.001, respectively.

## Data Availability

No new data were created or analyzed in this study. Data sharing is not applicable to this article.

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
