# Peer review of "Distinct Classes of Flavonoids and Epigallocatechin Gallate, Polyphenol Affects an Oncogenic Mutant p53 Protein, Cell Growth and Invasion in a TNBC Breast Cancer Cell Line"

_cells, 2021, doi:10.3390/cells10040797_

Round 1
Reviewer 1 Report
The authors tested several plant-derived compounds in triple-negative breast cancer cell lines with mutant p53. The experimental set-up is overly simple and not sufficient for the conclusion the authors have drawn from the study. Significant improvements would have to be made before resubmission.
Abstract should be carefully corrected for English language. The rest of the text should also be proof-read for small spelling/grammar corrections.
Introduction: the paragraph describing prior knowledge about each compound should be shortened.
General flaw in the results:
Statistical analysis was not done on any of the results! It seems that some of the experiments have been performed only once (no SD in quantification graphs). All experiments should be performed at least three times (even for WB) and this documentation should be made available to reviewers.
If stating that mtp53 is crucial for driving the effects of compounds then cells with wild type p53 have to be included in the study.
Which p53 mutations do the three cell lines have? Could that be the reason for different responses to compounds? This way, as authors stated in the discussion, other mutations (of many) could be responsible for the differences in the response.
Comments to Fig. 1:
Are the compounds causing specific cell cycle arrest or apoptosis? Cell cycle analysis after 72h of treatment might help decipher this (if more cells are in subG1). Apoptosis assays might also be used: Caspase-based viability assays or WB.
Comments to Fig. 2:
In the text referring to Fig. 2 there is a paragraph describing PCR results in Fig. 3. Please reorganize.
What is the right-hand graph in fig. 2B showing? It is not described in fig. legends.
If replication stress is induced by compounds, this should increase p53 protein level, however here we see a decrease of the mutant p53 (although wt p53 antibody was used). How do the authors explain this? What is then the mechanism of cell cycle arrest?
Comments to Fig. 3:
Some statements in the text do not correspond to what is shown in the graphs (e.g. ETS2 is not downregulated by EGCG). To simplify the reading, the significantly changed (or most relevant) genes can be presented in main figures and the rest shifted to supplementary figures. These results are descriptive and can’t be used to describe a mechanism. Specific gene(s) knock out or knock down would have to be done to support the mechanism proposed by the authors.
Discussion:
Repetition of results should be avoided.
The last paragraph in the discussion should be left out. Conclusions coming from purely in vitro data can not be extrapolated to human patient situation. Especially this conclusion is extremely overexaggerated: “Regular consumption of fruits and vegetables that are rich in these compounds may help prevent mtp53 TNBC cancers or at least significantly reduce cancer cell growth and metastasis.”
In vivo experiments would be necessary to imply such a conclusion.
With best regards!
Author Response
Reviewer 1:
The authors tested several plant-derived compounds in triple-negative breast cancer cell lines with mutant p53. The experimental set-up is overly simple and not sufficient for the conclusion the authors have drawn from the study. Significant improvements would have to be made before resubmission.
Response: Now new control data on WI38, a non-cancerous cell line containing wildtype p53 has been added. We introduced R249S mutation into WI38 and strikingly we could see upregulation of mutant p53 targets. Also treatment with these plant-derived compounds reduced NMG (nucleotide metabolism genes) much more profoundly compared to control WI38 cells. Also there is an increased cleaved PARP when we concurrently introduced R249S and treated it with these compounds. This new data to some extent is sufficient to draw some conclusions, for example these compounds have opposing effects on wildtype and mutant p53. This indicates these compounds are potentially less toxic with greater therapeutic window. Significant improvements have been made in the revised manuscript. I agree with the reviewer that this study was performed using simple techniques. Nonetheless for the first time we were able to show that we can downregulate a very highly stable mutant p53 protein using less toxic compounds. I believe this study is first of its kind. In the future we will aim to delineate the mechanisms behind the downregulation of mutant p53.
Abstract should be carefully corrected for English language. The rest of the text should also be proof-read for small spelling/grammar corrections.
Response: According to the reviewer’s suggestion, now the abstract and rest of the text was proof-read and corrected with the help of a native English speaker who is currently a cancer researcher.
Introduction: the paragraph describing prior knowledge about each compound should be shortened.
Response: I agree with the reviewers. Accordingly, I substantially reduced the description of prior literature.
General flaw in the results:
Statistical analysis was not done on any of the results! It seems that some of the experiments have been performed only once (no SD in quantification graphs). All experiments should be performed at least three times (even for WB) and this documentation should be made available to reviewers.
Response: Except western blotting and flow cytometry rest all experiments were done in three replicates. Now p-value is determined and showed significance. I agree with the reviewer that all experiments have to be replicated. I carried out this project in the USA and left the lab before submitting. Now it is more difficult to perform the repeats. I would like to stress that our westerns show that many nucleotide metabolism gene expressions were downregulated consistently, not just a few. I believe this data is not by chance that we achieved. Further when performing flow cytometry, I collected 10,000 events or cells in order to quantify the cell cycle phases. I believe this work is very timely and has a huge impact on mutant p53 cancers given the fact that there is not even a single drug that could decrease the expression of mutant p53. I strongly feel that this work should not be left out due to the lack of replicates for westerns. I hope the reviewer understands my concern. Moreover, it is very difficult to perform experiments due to the current worldwide situation.
If stating that mtp53 is crucial for driving the effects of compounds then cells with wild type p53 have to be included in the study.
Response: This is a very valid point. Now I included western blot data for a normal fibroblast cell line, WI38, harbouring wildtype p53 (Supplementary Figure 1).
Which p53 mutations do the three cell lines have? Could that be the reason for different responses to compounds? This way, as authors stated in the discussion, other mutations (of many) could be responsible for the differences in the response.
Response: The three cell lines used have different mutations. BT549, HCC38 and MDAMB231 have R249S, R273L and R280K p53 mutations, respectively. This was written in the manuscript. All these are hotspot mutations, meaning that they are highly stable and oncogenic. Especially R249S mutation harboured by BT549 (used primarily for this study) is highly oncogenic. Introduction of this mutation in a normal fibroblast cell line promoted high proliferation rate (Kollareddy et al., Nature Communications, 2015). Also now I included the data showing that introduction of R249S mutation upregulated the expression of several nucleotide metabolism genes and also its co-operative partner ETS2.
Differential response could be due to a variety of reasons, uptake of drugs is variable across the cell lines owing to the fact that cell lines express different levels of drug transporters. Some mutant p53 cell lines including MDAMB231 are inherently resistant to anticancer drugs. Especially R280K mutation in MDAMD231 has strong anti-apoptotic activity. This is supported by Bae et al., 2013, Genes & Genomics.
Comments to Fig. 1:
Are the compounds causing specific cell cycle arrest or apoptosis? Cell cycle analysis after 72h of treatment might help decipher this (if more cells are in subG1). Apoptosis assays might also be used: Caspase-based viability assays or WB.
Response: Almost all compounds caused cell cycle arrest after 24 - 48 hours, however after 72 hours all cells died. The IC50s of the compounds are >10 µM. Leaving them for 72h with concentrations greater than >10 µM will result in off-targeted effects and untargeted cell death. Hence cell cycle analysis and other experiments were performed for 24 hours treatment. Now an apoptotic marker, cleaved PARP, is included.
Comments to Fig. 2:
In the text referring to Fig. 2 there is a paragraph describing PCR results in Fig. 3. Please reorganize.
Response: This is reorganized now. I am very thankful to the reviewer for spotting this.
What is the right-hand graph in fig. 2B showing? It is not described in fig. legends.
Response: The right-hand graph in figure 2B is image J densitometry analysis, showing quantification of p53 and ETS2 protein levels after normalizing to loading control actin. Now the labelling also changed to make it clear. It was described in legends in the first submission.
If replication stress is induced by compounds, this should increase p53 protein level, however here we see a decrease of the mutant p53 (although wt p53 antibody was used). How do the authors explain this? What is then the mechanism of cell cycle arrest?
Response: Very very good point and keen observation by the reviewer. In response to the replication stress, generally p53 protein levels increase and induce cell cycle arrest or apoptosis depending on the severity of stress. Mutants p53(s) are highly stable proteins, that’s why we see huge levels of mutant p53(s) regardless of cell cycle phase. In other words, mutant p53(s) levels are not cell cycle regulated. We have shown this previously (Kollareddy et al., 2015 Nature Communications). We do not want levels of mutant p53 going up in response to new drugs, this would be bad for patients, because high levels can aggressively promote metastasis and chemoresistance. Our aim is to reduce oncogenic p53 mutants by using less toxic compounds. How the plant derived compounds reduce the mutant p53 protein levels is an open question and a moot point. From our data it is clear that EGCG and Genistein reduce mutant p53 both at RNA and protein level, whereas Silymarin and 3,3 Diino reduce only at protein level. Delineating the mechanism of these compounds with respect to their effects on mutant p53 RNA expression and protein levels is my future aim. As we explained nucleotide metabolism genes are involved in progression of S-phase by providing sufficient nucleotide pools. Reduction of the NMG protein levels means reduced nucleotides and hence cell cycle arrest. It seems that NMGs were reduced because its main regulator mutant p53 is reduced.
Comments to Fig. 3:
Some statements in the text do not correspond to what is shown in the graphs (e.g. ETS2 is not downregulated by EGCG). To simplify the reading, the significantly changed (or most relevant) genes can be presented in main figures and the rest shifted to supplementary figures. These results are descriptive and can’t be used to describe a mechanism. Specific gene(s) knock out or knock down would have to be done to support the mechanism proposed by the authors.
Response: I agree with the reviewer that EGCG is not downregulated significantly at the RNA level, although we see a bit of reduction at 1/2x IC50. Certainly, the ETS2 protein level is convincingly reduced in a dose dependent manner. I removed/amended the statements in the text which do not correspond to the data/graphs. As you suggested, I included only the most relevant genes in the main text and the rest of them showed in supplementary. I agree at some place the results are descriptive and should not be used to describe the mechanisms. I removed the descriptive parts and instead included in the discussion. Knockout and knockdown studies for most relevant genes will be performed in my future work. In relation to knockdowns, we previously showed that knockdown both stable and transient, substantially reduced the NMGs protein levels. Please see the data below. Additional data including reduction of nucleotide levels can be found in Kollareddy et al., 2015, Nature Communications. Also in new supplementary Figure 1 we knocked in R249S p53 mutation (BT549 p53 mutation) in WI38. Doing so readily increased the protein levels of several NMGs which are the targets of mutant p53.
Discussion:
Repetition of results should be avoided.
Response: Now I removed the text which reflects the repetition of results in discussion. Now I added few more sentences in discussion regarding the rationale of using these compounds. Thank you for your kind suggestions. This has been very useful in refining the manuscript.
The last paragraph in the discussion should be left out. Conclusions coming from purely in vitro data cannot be extrapolated to human patient situation. Especially this conclusion is extremely overexaggerated: “Regular consumption of fruits and vegetables that are rich in these compounds may help prevent mtp53 TNBC cancers or at least significantly reduce cancer cell growth and metastasis.”
In vivo experiments would be necessary to imply such a conclusion.
Response: Now I removed the last paragraph and I completely agree with the reviewer that in vitro that cannot be extended to the human patient situation without the in vivo data.

Reviewer 2 Report
The manuscript from M. Kollareddy and Luis A. Martinez titled “Distinct classes of flavonoids and Epigallocatechin gallate, a polyphenol, inhibits mutant p53 breast cancer proliferation and invasion through affecting expression of nucleotide metabolism genes” show some flavonoids tested reduced the expression of mutant p53 and its important targets that promote invasion and metastasis of breast cancer cell lines.
This research could have a good potential, however it has several methodological shortcomings, the main one being the absence of statistical analysis. In addition, control experiment should be better define.
Minor point: Fig 1B – what represent numbers on histograms?
Fig 1C- data on quercetin are missing
In short, I do not recommendate this work for publication in the present form.
Author Response
Reviewer 2:
The manuscript from M. Kollareddy and Luis A. Martinez titled “Distinct classes of flavonoids and Epigallocatechin gallate, a polyphenol, inhibits mutant p53 breast cancer proliferation and invasion through affecting expression of nucleotide metabolism genes” show some flavonoids tested reduced the expression of mutant p53 and its important targets that promote invasion and metastasis of breast cancer cell lines.
This research could have a good potential, however it has several methodological shortcomings, the main one being the absence of statistical analysis. In addition, control experiment should be better define.
Response: Now the p-value and statistical analysis is included and showed significance. Also I have new data in a control cell line WI38 (normal fibroblast) (supplementary Figure 1). This control experiment shows opposing effects, i.e these plant derived compounds reduced mutant p53 protein levels, whereas in a non-cancerous cell line, they increased wildtype p53 protein levels. This shows that these compounds are specific to mutant p53 cell lines. This indicates that they are less likely to be toxic to non-cancerous cells.
Minor point: Fig 1B – what represent numbers on histograms?
Response: Numbers on histograms represent doubling times in hours as deduced described in the methods. More the inhibition of cell proliferation rate, higher the doubling times.
Fig 1C- data on quercetin are missing
Response: I apologize for this. Thank you for spotting this. I included quercetin data now.
In short, I do not recommendate this work for publication in the present form.
Response: Based on the reviewers comments, I included new data, rewritten the manuscript (improved English with the help of the native speaker) and reorganized the manuscript. I took the reviewers comments positively and implemented the changes as much as I can. I believe that the revised manuscript is much improved and suitable for publication.
Reviewer 3 Report
The manuscript would benefit of a minimum of statistical analysis applied to the main results of your study. Basically, the Western Blotting and RealTime qPCR results need to be supported by determined levels of significance. As you mention in the text, paragraph 3.2: “Almost all the genes were downregulated significantly at 1 x IC50”; “…and IMPDH1 were significantly downregulated”; “…the reduction of ribonucleotides to deoxyribonucleotides) were downregulated significantly”, the level of significance is not well represented in the corresponding Figures, and in Material and Methods section you don’t describe the type of statistical test applied to the results (which allow your results to be "significant").
In particular, in the Figure 2 regarding the Western Blotting analyses, it should be included a quantitative graphic (bar charts) with standard deviation and significance among the tested targets. At the moment, it’s only presented the densitometric analysis of the p53 protein levels. This only partially follows the text (paragraph 3.2.), in which you state that “Almost all the genes (or proteins?) were downregulated “significantly” ...”.
A similar concept would be appliable for the qRT-PCR data, represented as bar charts but without any significance represented on. Should I assume that it is not significant at all, or that it simply lacks of a statistical analysis?
Minor comments:
Abstract: “50% of all cancer p53 mutations and correlates” - Unclear sentence
Introduction:
- “Few drugs have shown to be effective in restoring wtp53 activity” you mean wild-type p53 right?
- The Introduction needs to be revised, it seems rather a discussion and it poorly introduces the study. You may choose to synthetize the body of this paragraph, it’s too long (one or two sentences for each compound are sufficient in this section).
Material and Methods:
- Cell proliferation assay: 10,000 cells? Were they seeded in 96 well-plates?
- qRT-PCR: please specify which internal reference/housekeeping gene/s you used in the analyses.
- Please add a statistical analysis sub-section.
Results: “3.3. Natural compounds downregulated transcripts of mtp53 and its targets” This title is very similar to the previous one “3.2. Natural compounds downregulated mtp53 and its direct transcriptional targets”.
Discussion: “Metastasis of malignant cells to distant organs is one the main causes of mortality. Inhibition of metastasis by chemopreventive approaches would be useful to reduce mortality rates.” I think these sentences can be removed.
Author Response
Reviewer 3:
The manuscript would benefit of a minimum of statistical analysis applied to the main results of your study. Basically, the Western Blotting and RealTime qPCR results need to be supported by determined levels of significance. As you mention in the text, paragraph 3.2: “Almost all the genes were downregulated significantly at 1 x IC50”; “…and IMPDH1 were significantly downregulated”; “…the reduction of ribonucleotides to deoxyribonucleotides) were downregulated significantly”, the level of significance is not well represented in the corresponding Figures, and in Material and Methods section you don’t describe the type of statistical test applied to the results (which allow your results to be "significant").
In particular, in the Figure 2 regarding the Western Blotting analyses, it should be included a quantitative graphic (bar charts) with standard deviation and significance among the tested targets. At the moment, it’s only presented the densitometric analysis of the p53 protein levels. This only partially follows the text (paragraph 3.2.), in which you state that “Almost all the genes (or proteins?) were downregulated “significantly” ...”.
A similar concept would be appliable for the qRT-PCR data, represented as bar charts but without any significance represented on. Should I assume that it is not significant at all, or that it simply lacks of a statistical analysis?
Response: Now the level of statistical significance is included for RT-PCR and invasion assays. Also the statistical method applied to deduce p-value is also written. I could not do the same for western and flow cytometry. Except western blotting and flow cytometry rest all experiments were done in three replicates. I carried out this project in the USA and left the lab before submitting. Now it is more difficult to perform the repeats for western and flow cytometry. I would like to stress that our westerns show that many nucleotide metabolism gene expressions were downregulated consistently, not just a few. I believe this data is not by chance that we achieved. Further when performing flow cytometry, I collected 10,000 events or cells in order to quantify the cell cycle phases. I believe this work is very timely and has a huge impact on mutant p53 cancers given the fact that there is not even a single drug that could decrease the expression of mutant p53. I strongly feel that this work should not be left out due to the lack of replicates for westerns. I hope the reviewer understands my concern. Moreover, it is very difficult to perform experiments due to the current worldwide situation.
Does the reviewer mean to show densitometric quantification data for all proteins? I quantified only p53 and ETS2 by densitometric analysis as these two are critical transcriptional regulators of NMGs.
Minor comments:
Abstract: “50% of all cancer p53 mutations and correlates” - Unclear sentence
Response: I agree that this sentence is unclear. I have rewritten. I would like to thank you for correcting this.
Introduction:
- “Few drugs have shown to be effective in restoring wtp53 activity” you mean wild-type p53 right?
- The Introduction needs to be revised, it seems rather a discussion and it poorly introduces the study. You may choose to synthetize the body of this paragraph, it’s too long (one or two sentences for each compound are sufficient in this section).
Response: I mean that few compounds were able to bind to mutant p53 and change its conformation to wildtype p53 and partially restore wildtype activity (more apoptosis in response to DNA damaging agents). Yes I mean wildtype. I revised the introduction and wrote only a few sentences for each compound. I agree that the introduction to some extent seems like a discussion. I have amended the introduction based on your suggestion.
Material and Methods:
- Cell proliferation assay: 10,000 cells? Were they seeded in 96 well-plates?
- qRT-PCR: please specify which internal reference/housekeeping gene/s you used in the analyses.
- Please add a statistical analysis sub-section.
Responses: Yes they were all seeded in 96 well plates. I have updated now. This is a three-day assay and I previously optimized the cell number. All three cell lines have similar proliferation rates. I have used HPRT as a housekeeping gene for normalization. Now I updated in the methods. Now I added a statistical analysis sub-section in the methods.
Results: “3.3. Natural compounds downregulated transcripts of mtp53 and its targets” This title is very similar to the previous one “3.2. Natural compounds downregulated mtp53 and its direct transcriptional targets”.
Responses: I have changed the title of this sub-heading to avoid confusion. This is a very valuable suggestion.
Discussion: “Metastasis of malignant cells to distant organs is one the main causes of mortality. Inhibition of metastasis by chemopreventive approaches would be useful to reduce mortality rates.” I think these sentences can be removed.
Responses: I removed these sentences as we need in vivo data to support such kinds of statements.
Round 2
Reviewer 1 Report
The authors responded to all comments, therefore I recommend the article to be accepted in present form.
Only one more suggestion: if there is a viability assay done with WI38 cells, it would be good to add it to suppl. fig. 1.
All the best!